# RIFT Process Analysis for the Production of Green Composites in Flax Fibers and Bio-Based Epoxy Resin

**DOI:** 10.3390/ma15228173

**Published:** 2022-11-17

**Authors:** Luca Sorrentino, Sandro Turchetta, Gianluca Parodo, Roberta Papa, Elisa Toto, Maria Gabriella Santonicola, Susanna Laurenzi

**Affiliations:** 1Department of Civil and Mechanical Engineering, University of Cassino and Southern Lazio, 03043 Cassino, Italy; 2Department of Chemical Engineering Materials Environment, Sapienza University of Rome, Via del Castro Laurenziano 7, 00161 Rome, Italy; 3Department of Astronautical Electrical and Energy Engineering, Sapienza University of Rome, Via Salaria 851-881, 00138 Rome, Italy

**Keywords:** green composites, vacuum infusion process, permeability, compressibility, finite element analysis

## Abstract

In this work, a dual objective is carried out on composite materials in flax fiber and bio-based epoxy resin: to determine the process parameters and to develop a numerical model for highlighting the potential of and the limits in the production of “green” laminates through a RIFT process (Resin Infusion under Flexible Tool). For these reasons, compressibility tests were performed in order to evaluate the behavior of commercial flax woven under the vacuum bag. Subsequently, permeability tests were performed in order to evaluate the permeability curves necessary for the numerical study of the infusion process. For the numerical analyses, the commercial software PAM-RTM was adopted and validated. In this work, vaseline oil was used as the injected resin for the validation, and a bio-based epoxy commercial system was used for the study of the infusion process in a simple case study. The results were compared with a petroleum-based epoxy system typically used for infusion processes, showing the potentiality and the critical use of bio-based resins for infusion processes.

## 1. Introduction

In recent years, due to the continuous debates on the problems related to environmental impact and the demands for new and more stringent regulations, companies need renewable resources to make environmental protection one of their priorities and aim to achieve environmental sustainability objectives. In particular, in the road transport sector, the demand for increasingly lighter vehicles capable of maximizing performance and solving the problems associated with CO_2_ emissions is growing. In this perspective, the use of composite materials is constantly growing, in particular those “green” materials that have characteristics of biodegradability and recyclability. Natural fibers are used as reinforcement materials as an alternative to glass, but they are not able to guarantee the stiffness and mechanical strength of parts made with carbon fibers. Nowadays, green composite materials can represent a compromise between sustainability and good performance. Generally, parts made with green composites are produced through Liquid Composite Molding processes (LCM) [1,2,3].

While the scientific literature on the infusion process of synthetic fibers provides a comprehensive set of studies on the methodology of measurements, impregnation, and simulation of the process, in the last decade, the study of the infusion of reinforcements in natural fibers has found a growing interest because of their characteristics of biodegradability, low energy consumption for production, and consequently low environmental impact [4,5]. In fact, it is estimated that the production of flax fibers leads to a lower energy consumption of about 82% compared to the production of glass fibers [6].

Compared to synthetic fibers, natural fibers differ in morphology, structure, and chemical composition, which influence the filling of the cavity during the infusion process. The main difficulty for the infusion of natural fiber parts consists in evaluating the compressibility and permeability of the reinforcement. Regarding the morphology, the natural fibers are not full (as is the case for synthetic fibers) but have a central cavity, called a lumen, whose size also depends on the manufacturing process and the adopted plant [7]. Generally, flax and hemp fibers showed little of the lumen dimension with respect to other fibers, such as wood and kenaf [8], and this is why the fibers obtained from such plants have seen a growing interest in their application to the study of realizing parts in a green composite material. In particular, flax fibers have a polygonal section structure, which theoretically allows them to reach a volumetric fiber fraction higher than those of glass (which have a circular section). However, today’s technological processes do not allow for the optimal positioning of the reinforcement, resulting in a real fiber fraction lower than the theoretical one. In addition, LCM processes provide for the compaction of the reinforcement that, in the case of natural fibers, can lead to failure of the fiber structure with the closure of the lumen. This results in an irreversible deformation of the natural reinforcement that is not present in synthetic reinforcements [9].

Usually, the models used for the study of the infusion process are based on Darcy’s law [4,10,11], in which the characteristic parameter of the impregnation capacity of the reinforcement is represented by the permeability. In fact, the study of the resin flow inside the reinforcement allows the prediction of any defects (such as voids, dry spots, or excessive porosity) that can be generated inside the manufactured part, leading to lower mechanical performance [5,12,13,14]. Developments of such models include the implementation of thermo-chemical models, which take into consideration cure kinetics and thermal fluxes during the process [15] in order to evaluate possible non-uniformity of the degree of cure in the part and the presence of deleterious exothermic peaks [16]. For this purpose, it is important to determine the conductive coefficients of the laminate, which can generally be determined not only experimentally but also using homogenization software (such as Digimat, version 2018.1), which allows the determination of the macroscopic properties of the layer starting from its individual constituents [17,18,19]. However, the use of this approach for the study of the infusion of natural fiber parts could be influenced by the lack of homogeneity in the reinforcement properties.

Elinor Swery et al. [11,20] highlighted two measurement methods to calculate in-plane and through-thickness permeabilities. In-plane permeability tests can be carried out on unsaturated and saturated samples using a radial flow injection scheme. Adding a dye to the used fluid, they evaluated the unsaturated–saturated permeability ratio. R. Umer et al. [12,21] illustrated the effect of flax yarn size on the permeability and compaction of the mat. In particular, they noted that permeability is reduced by increasing the diameter of the yarn and decreasing the length of the fibers. This was due essentially to the difference in the twist for each strand of yarn and, subsequently, in different surface structures of the yarn. R. Mbakop et al. [13,22] studied the effect of compaction on permeability as a function of temperature and humidity in the UD flax-mat reinforcement. They observed an increase in the permeability and mechanical strength of tensile test specimens made using the RTM process. In addition, the elastic modulus was not affected by the reinforcement processing state. G. Francucci et al. [14,23] replaced glass fibers with natural fibers; in particular, jute fabric was characterized in terms of saturated and unsaturated permeability. The higher the absorption was, the lower the flow rate was. In particular, jute fibers showed a higher absorption rate than glass fibers, leading to a lower flow rate. This aspect can cause swelling, reduced porosity, and increased flow resistance for the infusion of jute fibers. Similar results have been observed in [15,24].

In most of the previous studies, the focus was on the analysis of permeability and characterization of natural fibers for infusion processes. However, the investigated resins were usually petroleum-based, limiting the potentiality of sustainability for this type of material. Accordingly, in this work, the processability of “green” composite materials manufactured with natural fibers and bio-based resins has been experimentally analyzed, highlighting the potential and limitations in the use of bio-based resins combined with woven-in natural fibers.

The objective of this work is to study the limits found in the product/process design of natural fibers and bio-based resin components through the RIFT (Resin Infusion under Flexible Tool) process. To ensure the analysis of the numerical model (with the help of the PAM-RTM software, version 2020.5), it is necessary to carry out permeability and compressibility tests in order to obtain numerical parameters that are fundamental for the study of the process for which—for the materials used—neither literature nor data sheets are currently available. In particular, in this paper, experimental tests have been carried out to measure the permeability and compressibility of a woven flax fabric as a function of the fiber fraction. The tests carried out were also used to validate the numerical model.

## 2. Materials and Methods

The reinforcement used in this work was a twill 2 × 2 flax fabric (ampliTex 5040, manufactured by Bcomp, Fribourg, Switzerland). The fabric specifications are reported in Table 1. Regarding the investigated bio-based resin, an AMPRO BIO (manufactured by Gurit Ltd., Wattwil, Switzerland) [25] epoxy multipurpose system was adopted. The choice of this epoxy system was due to the elevated bio-based carbon content of the product (up to 60%) and the versatility of this system for other applications, such as bonding, laminating, and filling. Moreover, this epoxy system has been compared with a typical infusion system known as PRIME 20LV (also manufactured by Gurit Ltd., Wattwil, Switzerland) [26]. The characteristics of the two epoxy systems are reported in Table 2.

### 2.1. Compressibility Behaviour

Compressibility tests (Figure 1) were carried out in order to obtain the relationship between the pressure to which the laminate was subjected and its volumetric percentage of fibers. In fact, the volumetric fraction of fibers can be defined as:(1)$$V_{f}=\frac{n\;A_{f}}{h\;\rho }$$
where $V_{f}$ is the volumetric fiber content, n the number of layers, $A_{f}$ the areal density of the fabric, h the thickness, and ρ the density of the fibers. For the tests, a universal testing machine equipped with compression plates of 100 mm × 100 mm size and a load cell of 10 kN was used. The tests were carried out with a crosshead speed of 2 mm/min. To carry out the tests, the fabric was cut with dimensions equal to the dimensions of the compression plates.

Compressibility tests were realized using laminates made up of 10 layers. For compressibility tests, five replicates were performed for each condition. From the compressibility tests, it was possible to obtain the load-displacement curves and thus the variation in distance between the compression plates, which represented the difference in thickness ∆*h*:(2)$$∆h=h-h_{0}$$
where h was the final thickness and $h_{0}$ the initial thickness of the laminate.

### 2.2. Permeability Behaviour

Permeability tests (Figure 2) were realized on laminates made up of twill 2 × 2 flax fabric with the dimensions of 200 mm × 150 mm.

The test fluid used was a vaseline oil with a viscosity of 0.110 Pa‧s and a relative density of about 0.860.

The test fluid was injected laterally into the experimental equipment and placed under straight flow and constant volume flow rate conditions. In the case of straight flow, Darcy’s law is written as follows:(3)$$\dot{Q}=-\frac{A\cdot K\cdot ∆P}{\mu \cdot L}$$
where Q was the volume flow rate, *A* was the cross-section, K was the permeability of the porous medium, ∆P the pressure gradient, *μ* the viscosity coefficient, and *L* the length of the specimen.

The effective cross-section $A^{\prime }$ can be defined as a function of the porosity of the laminate (φ):(4)A′=A·φ

After combining Equations (3) and (4), the volume flow rate was equal to:(5)$$\dot{Q}=\vec{v\;}\cdot A\prime$$
where $\vec{v}$ represented the flow front velocity. In this way, it was possible to define the permeability through Equation (6):(6)$$K=\frac{\vec{v\;}\cdot \varphi \cdot L\cdot \mu }{∆P}$$

In order to determine the permeability curve as a function of the volumetric fraction of fibers, tests were carried out on the laminates by varying the number of layers. The test bench used consisted of a rectangular aluminum base of the dimensions 400 mm × 400 mm × 10 mm and a counter-mold made of plexiglass of the dimensions 240 mm × 330 mm × 10 mm, which allowed having a visual control of the flow front advancement in the laminate. Spacers separated by a distance of 130 mm were used to fix the height of the cavity. The fabrics, cut with the same dimensions as the cavity, were placed on the lower mold.

The resin inlet was positioned on the left of the mold, while the vacuum port was positioned on the right. Moreover, a transparent grid was positioned on the counter-mold in order to make a visual inspection of the flow front advancement during the test.

For the tests, 3.2 mm thick gauges were used, and the number of plies was varied in order to obtain a volumetric percentage of fibers variable for each test condition. Knowing the volumetric percentage of fibers, the volumetric flow rate of the fluid and its viscosity, and the difference of pressure between the inlet and outlet and the distance traveled by the fluid, it was possible to derive the permeability value along the principal direction by applying Darcy’s law.

For permeability tests, three levels of fiber volumetric fraction were considered, namely 0.14, 0.35, and 0.55. Moreover, five replications were made for each level.

For the experimental measurement of the flow front during impregnation, a video camera was used to record the progress as a function of time. The measurements were taken using reference markers positioned above the glass counter-mold and spaced 20 mm apart. From the experimental data collected and the application of Darcy’s law, it was possible to derive permeability for each test.

### 2.3. Resin Kinetic Behavior

Thermal analysis of the resin was performed with differential scanning calorimetry using a DSC 8500 (PerkinElmer). The instrument was calibrated with high-purity indium and tin materials. Experiments were carried out under a nitrogen flow of 40 mL/min on samples of 5–8 mg sealed in aluminum pans with lids. An identical empty cell was taken as a reference. Five samples were analyzed to ensure the reproducibility of results.

The curing behavior of the resin was investigated under non-isothermal and isothermal conditions. DSC samples were prepared immediately after the addition of the hardener to the resin base. Dynamic analysis was performed at heating rates of 5, 10, 15, and 20 °C/min, in the temperature range from 15 to 200 °C. The exothermic peaks were analyzed using the thermal analysis software provided with the instrument. The PerkinElmer Kinetics Software package (version 13.3.1.0014) was used to process the heat flow data, obtaining the extent of conversion (α) as a function of temperature. DSC static analysis was performed according to the ASTM E 2070 standard, and Test Method A was used to establish the kinetic parameters of the curing reaction [16,27]. Isothermal experiments were performed at temperatures of 40, 43, 47, and 50 °C. These temperatures were selected to be between 1 and 10% of the total reaction following the ASTM standard. Test Method A allows for determining activation energy, pre-exponential factor, and reaction order using differential scanning calorimetry from a series of isothermal experiments over a small (10 K) temperature range. This treatment is applicable to low *n*th order reactions and to autocatalyzed reactions such as thermoset curing.

The curing analysis was based on the assumption of proportionality between the heat of the reaction completed (Δ*H_c_*) and the fraction converted (*α*), as reported in the following equation:(7)$$\alpha =\frac{\Delta H_{c}}{\Delta H}$$
where Δ*H* is the total heat of the reaction. Starting from the dynamic DSC analysis of the resin samples, the extent of conversion was plotted as a function of temperature at each fixed scan rate. For each isothermal curve, a linear baseline was determined between the beginning and the end of the curing peak. The heat of the reaction Δ*H* was calculated by integrating the total area of the peak bounded by the peak itself and the baseline. After identifying the times corresponding to 10 and 90% of the peak area, ten equally spaced time values were identified between these limits. The rate of reaction (*dH*/*dt*) and the heat of the reaction completed (Δ*H_c_*) were calculated for each interval, and the fractional rate of reaction (*dα*/*dt*) was obtained using the following equation: (8)$$\frac{d\alpha }{dt}=\left(\frac{dH}{dt}\right)/\Delta H$$

Autocatalyzed reactions, typical of thermoset curing, can be described by Equation (7) and cast in its logarithmic form (Equation (8)): (9)$$\frac{d\alpha }{dt}=k\left(T\right)\;\alpha^{m}{\left(1-\alpha \right)}^{n}$$
(10)$$\ln\left[\frac{d\alpha }{dt}\right]=\ln\left[k\left(T\right)\right]+m\ln\left[\alpha \right]+n\ln\left[1-\alpha \right]$$

Equation (8) was solved using multiple linear regression analysis. The change of *k*(*T*) with temperature is described by the Arrhenius equation:(11)$$k\left(T\right)=Z\;e^{-E/RT}$$
where *Z* is the pre-exponential factor, *E* is the activation energy, *R* is the gas constant, and *T* is the absolute temperature. The plot of *ln*[*k*(*T*)] versus *1*/*T* is linear, with the slope equal to −*E*/*R* and the intercept equal to *ln*[*Z*].

### 2.4. Numerical Model

The permeability tests were simulated in order to validate the numerical model. For numerical simulations, the PAM-RTM software (version 2020.5) developed by the Esi Group was used. The model implemented in the software was based on Darcy’s law:(12)$$\vec{V}=-\frac{k}{\mu }\;\vec{\nabla P}$$
where $\vec{V}$ was the velocity of the flux, k was the permeability, μ the viscosity of the fluid, and $\vec{\nabla P}$ the pressure gradient.

For the definition of the element size, a sensitivity analysis to the mesh was made: the simulations were realized using different numbers of elements for the entire domain (2184, 15,366, 61,482, and 121,415 elements, respectively). As a result, the mesh adopted for the simulation is shown in Figure 3 and consisted of 61,482 2D triangular elements, which were the optimal value obtained from the sensitivity analysis. The mesh dimensions were 130 mm × 200 mm, which were the nominal dimensions of the laminates for the permeability test.

As shown in Figure 3, the boundary conditions consisted of a fixed flux on the nodes of the left side of the mesh and a vent on the right side of the mesh with an imposed pressure equal to zero.

For modeling the possible cure of the matrix, the autocatalytic relations presented in the previous paragraph were considered. 

## 3. Results and Discussions

### 3.1. Compressibility Experimental Results

The average compressibility curve resulting from the experimental tests for flax woven is shown in Figure 4. Considering the five replicates, the dispersion of the results is always less than 17%. As expected, the fiber fraction increased with an increase in pressure. The flax reinforcement being a valid and sustainable substitute for the glass fibers, a comparison of the compressibility behavior is also reported in Figure 4. In particular, woven flax showed a similar compressibility behavior with respect to woven glass; nevertheless, the obtained fiber fraction was even lower [17,28]. Compressibility curves could be interpolated using a power law:(13)$$p=\theta \;V_{f}{}^{\xi }$$
where *p* was the pressure, while *θ* and ξ were material parameters to be implemented in the PAM-RTM software (version 2020.5) for numerical analyses. The obtained values are reported in Table 3, while in Figure 4, the interpolation curves are also reported (dotted line). As a result of the analysis of these fabrics, the maximal theoretical value obtainable for the fiber fraction through infusion processes of a composite part is 0.41 for flax and 0.54 for glass. This result has an important influence on the mechanical performance of the produced part and the filling time, as discussed in the next paragraph.

### 3.2. Permeability: Experimental and Numerical Results

The experimental results related to the permeability tests are shown in Figure 5. As expected, the permeability decreased as the fiber fraction increased. The experimental measurements could be interpolated using Equation (14):(14)$$K\left(V_{f}\right)=A\;exp^{B\cdot V_{f}}$$
where *A* and *B* were coefficients that depended on the reinforcement used. The obtained experimental values for *A* and *B* were respectively 1.6 × 10^−8^ and 1.6 × 10^−6.99^. These results were quite a bit lower than the glass woven permeability results reported in the scientific literature [11,18,19,20,29,30].

For this type of fiber, no parabolic flow front has been observed, except for a little variation between a light parabolic profile and a flat flow profile. This result is in accordance with the scientific literature [1]. This phenomenon was probably due to the swelling of the fibers in saturated regions, which allowed the formation of a flat profile in the unsaturated region during tests. The tendency to make a flat flow front was also due to the low velocity in filling the cavity.

The use of a bio-based resin, combined with lower permeability in the reinforcement, can represent a technological issue. In fact, generally, bio-based resins present a higher viscosity with respect to traditional infusion resin systems, which could not allow the filling of the cavity within the time limits set essentially by the pot life of the resin system. The use of solvents during resin mixing could mitigate this problem; however, nowadays, there are no in-depth studies to assess the short and long-term effects of this method on components made of “green” composites.

The lower values of reinforcement content, which can be reached by applying the vacuum during the RIFT process, on the one hand, facilitate the flow of resin inside the cavity while, on the other hand, reducing the mechanical performance of the green composite component compared to the same component made with glass fibers. However, the latter aspect can be partly mitigated by the fact that the density of flax fibers is lower than the density of glass fibers, so in terms of specific strength, the two types of reinforcement results are similar.

Permeability tests were numerically simulated in order to validate the model. The numerical model did not show the behavior of the flow front at a mesoscale, but it forecasted the behavior of the flow front at a macro scale. In addition to the permeability parameters of the material, the measured experimental vaseline oil flow values were implemented as an input boundary condition. In this way, it was possible to carry out simulations and compare the speed of the advancement of the oil flow inside the cavity. The comparison between numerical and experimental filling times is shown in Table 4, while, by way of example, Figure 6 reports the comparison between the front of advancement of the oil for three instants of time, relative to the permeability test with a fiber fraction equal to 0.55. It is possible to say that the model can predict with good accuracy the times for filling the cavity. This aspect is fundamental for the subsequent infusion analysis of the catalyzed resin, as the cavity itself must be filled within the resin processing window, which is well represented by its pot-life. If this were not achieved, the resin would reticulate, preventing the flow advancement and avoiding the complete filling of the mold. This aspect is of fundamental importance in the case of composites made with bio-based resins. In the following paragraphs, we will explain the kinetic parameters of the bio-based resin used in this work and the numerical filling times in a simplified case study.

### 3.3. Resin Kinetic Results

The curing behavior of the resin was analyzed starting from DSC dynamic and isothermal scans. Figure 7a shows typical DSC thermograms obtained for the resin at different heating rates (5, 10, 15, and 20 °C). Starting from the dynamic data, the extent of conversion was plotted as a function of temperature at each fixed scan rate, obtaining the curves reported in Figure 7b. The curves show a sigmoidal shape, which is typical of an autocatalytic reaction mechanism [20,31].

Isothermal test temperatures were identified to be those between 1 and 10% of the total reaction [16,27] and were found at 40, 43, 47, and 50 °C. An analysis of the isothermal scans was performed, taking into account the autocatalytic reaction mechanism as determined in the previous dynamic experiments. Equations (8) and (9) were used to establish the kinetic parameters of the curing reaction according to the ASTM E 2070 standard. Results obtained by applying Test Method A are reported in Table 5.

## 4. Future Development: RIFT Process Design

The obtained experimental and numerical results allowed the modeling of the infusion process for parts made of flax fabric and bio-based resin. In this work, a simplified case study is presented that allows us to highlight the main problems in the production of parts made with this type of material.

The case study consisted, essentially, of a flat plate of the same size as that analyzed in the previous paragraph, composed of 4 layers of woven flax, discretized through a 2D shell-type mesh with 60,678 triangular-type elements and characterized by the application of a hole in the center for resin injection (Figure 8). The injection port was defined differently from the permeability study in order to reduce and optimize the path that the bio-resin would have to make during the infusion.

The numerical procedure was similar to that reported for the vaseline oil, but this time the properties of the fluid would be replaced with those of the resin AMPROBIO obtained from kinetic analysis. In order to consider the cross-linking of the resin, the kinetic parameters have been implemented in the software, and a tool temperature of 25 °C is set. In this case study, since heating systems were not considered, heat exchange with the external environment was not considered. As boundary conditions, a resin injection pressure of 100 kPa was considered (i.e., ambient pressure) for the injection port represented by the hole positioned in the center of the plate and a null pressure to the vent port represented by the perimeter of the plate. Moreover, a commercial petroleum-based epoxy resin for infusion Epoxy Prime 20LV (manufactured by Gurit UK Ltd.), was considered for comparison with the bio-based resin investigated in this work. The choice of this epoxy system was due to its compatibility with natural reinforcements, while its properties were taken from the datasheet and literature [3,32]. The viscosity and cure parameters of the Prime 20 LV resin were taken from the technical datasheet and literature [3,21,32,33].

The numerical results of the case study showed very different filling times. In the case of the vaseline oil, the filling of the cavity took place in 25.8 s; for the Epoxy Prime 20LV resin, the filling took place in 53.9 s; while for the biobased resin AMPROBIO, the filling took place in 229.8 s (Figure 9). This difference in time of filling between the petroleum-based resin and bio-based resin represented an important limit for the infusion of “green” composites using bio-based resins. In fact, since the time required for a complete cure was similar in each case and about equal to half an hour, the difference in the filling times for the cavity was crucial for the success of the infusion itself.

The analysis of the numerical results also showed that the variation in the degree of cure during the filling of the cavity was negligible for both epoxy systems. This is mainly due to the small size of the case study and the process temperature being set to 25 °C (ambient temperature). However, in the case of larger parts, the longer filling time for the bio-based resin led to a growth in the degree of cure of the resin and, subsequently, to an increase in its viscosity. The adoption of process temperatures above room temperature could reduce the filling time on the one hand and increase the resin cure rate on the other. Starting from these observations, the studies of the process of infusion of bio-based resins and product/process optimization methodologies will be the subject of subsequent work by the authors.

## 5. Conclusions

The bibliographical introduction of green composite materials has highlighted the presence of several works focused on the use of natural fibers but, at the same time, has highlighted the lack of work on the use of organic resins to ensure complete environmental sustainability. The present work focuses on the use of both fibers, and natural and organic resins that make a composite material “green” and in which parameters such as the permeability and compressibility of fabrics are fundamental for the development of components useful to different industrial sectors. In this regard, permeability and compressibility tests were conducted on woven flax using vaseline oil as the test fluid. The experimental tests allowed us to obtain the constitutive parameters of the material and to validate the model implemented in the PAM-RTM software (version 2020.5).

The next step consisted of the characterization of the bio-based resin, thus obtaining the kinetic parameters necessary for the numerical study of the curing process. These parameters were used for the study of a simple case that highlighted the main limitations of bio-resin compared to an equivalent petroleum-based resin. In fact, the cavity filling times for the bio-resin were found to be more than four times higher than those of the competing petroleum-based resin. The small size of the simple case study showed no problems related to an increase in the degree of cure during the filling phase. However, this aspect needs further study as it appears to be of fundamental importance in the case of the infusion of larger complex parts. In fact, this work lays the foundation for the future development of components for various industrial sectors in order to project the production towards an ecological and sustainable footprint for the environment.

## Figures and Tables

**Figure 1 materials-15-08173-f001:**
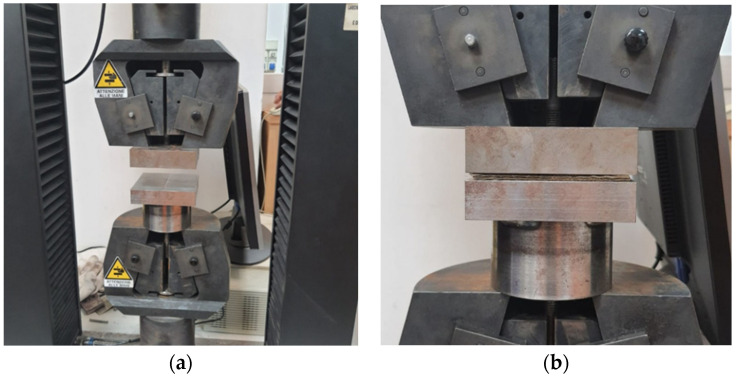
Experimental compressibility test: (**a**) universal testing machine equipped with the compression plates; (**b**) a particular specimen during the compression test.

**Figure 2 materials-15-08173-f002:**
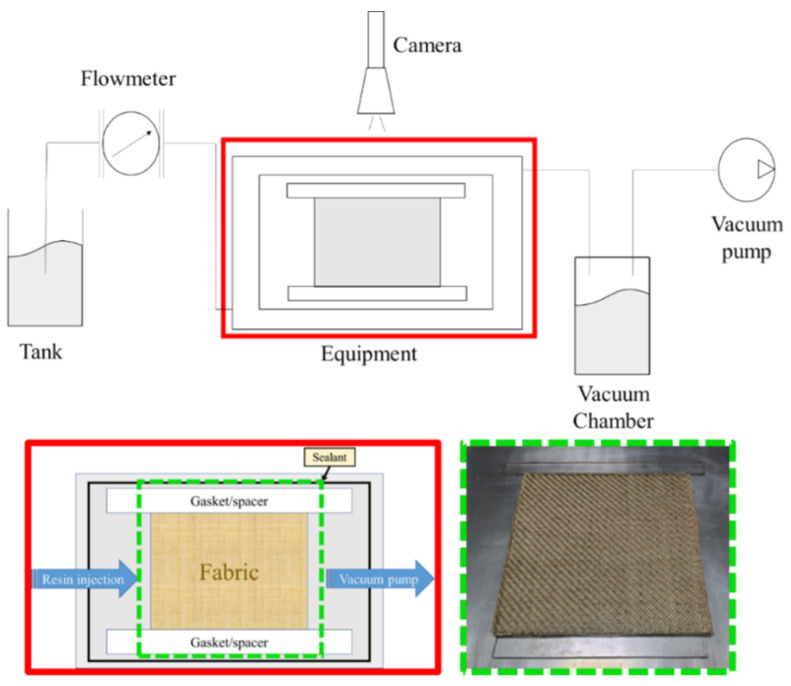
Experimental set-up used for the permeability tests.

**Figure 3 materials-15-08173-f003:**
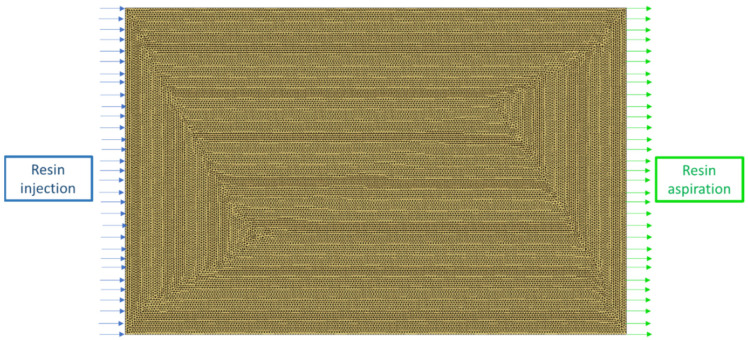
Mesh used for the permeability simulations and related boundary conditions.

**Figure 4 materials-15-08173-f004:**
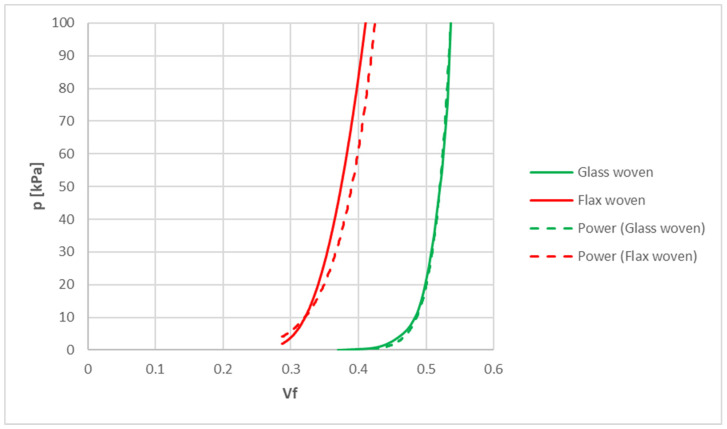
Experimental average compressibility curves comparison between flax woven (red) and glass woven (green).

**Figure 5 materials-15-08173-f005:**
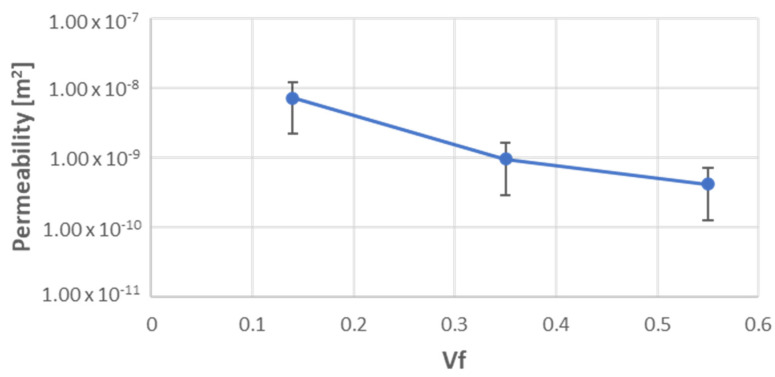
Experimental results related to permeability tests of woven flax.

**Figure 6 materials-15-08173-f006:**
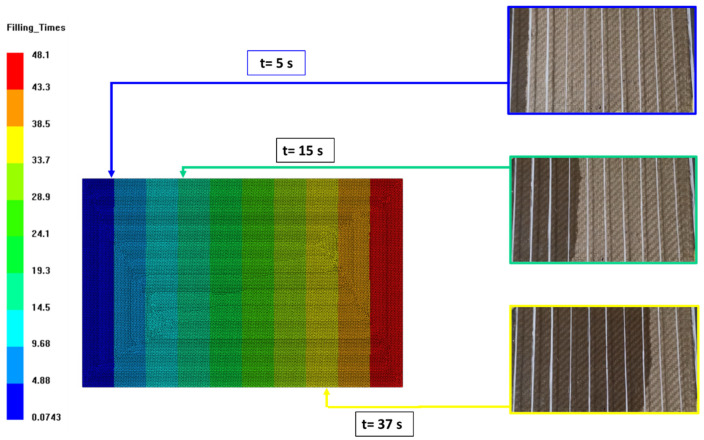
Comparison between numerical and experimental flow advancement fronts for three instants of the permeability test with a fiber fraction of 0.55.

**Figure 7 materials-15-08173-f007:**
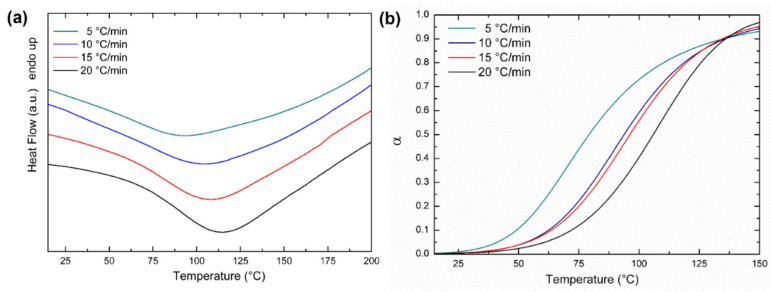
(**a**) DSC thermograms of resin at different heating rates. Data are offset for clarity; (**b**) Extent of conversion (α) as a function of temperature at different heating rates.

**Figure 8 materials-15-08173-f008:**
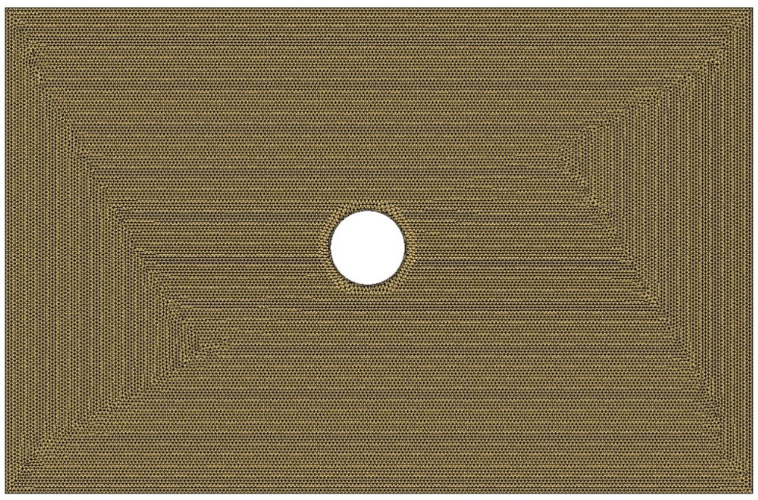
Mesh of the case study for resin infusion analysis.

**Figure 9 materials-15-08173-f009:**
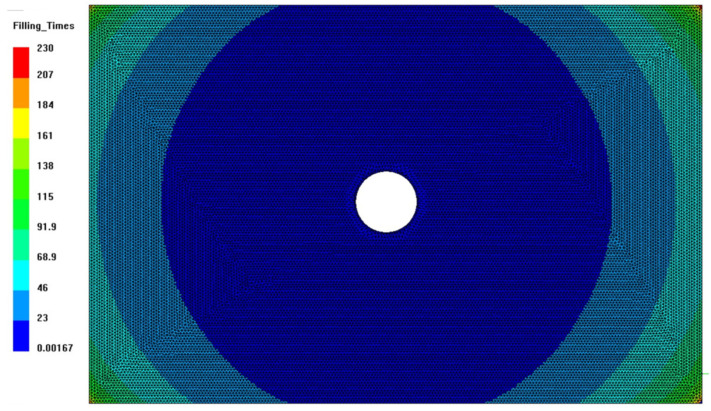
Filling time for AMPROBIO resin: the results of PRIME 20 LV resin were qualitatively similar, with a filling four times lower.

**Table 1 materials-15-08173-t001:** AmpliTex 5040 specifications.

Parameter	Value
Fiber density	1350 kg/m^3^
Elongation at break	1.4%
Tensile modulus	62 GPa
Tensile strength	224 MPa
Areal Weight	0.3 kg/m^2^
Weave	Twill 2 × 2

**Table 2 materials-15-08173-t002:** Characteristics of the investigated bio-based epoxy system AMPRO BIO and comparison with a petroleum-based system PRIME 20LV.

Parameter	AMPRO BIO	PRIME 20LV
Mixed viscosity [mPa s]	1100	230
Geltime at 25 °C [min]	45	30
Cured density [g cm^−3^]	1.09	1.15
Tensile modulus [GPa]	1.9	3.2
Tensile strength [MPa]	36.3	75.0
Failure strain [%]	>12.0%	4.1%

**Table 3 materials-15-08173-t003:** Compressibility parameters comparison between flax woven and glass woven.

Woven Material	*θ*	ξ	R-Squared Value
Flax	1.15 × 10^8^	8.229	0.949
Glass	1.00 × 10^9^	19.033	0.994

**Table 4 materials-15-08173-t004:** Experimental-numerical comparison of the filling times of the cavity.

Fiber Fraction	Experimental Average Filling Time [s]	Numerical Filling Time [s]	Difference from Experiments
0.14	4.1	4.2	2.4%
0.35	30.3	29.9	−1.3%
0.55	45.6	47.8	4.8%

**Table 5 materials-15-08173-t005:** Kinetic parameters of the resin curing reaction obtained from DSC isothermal analysis.

T (K)	ln [k(T)] (ln[min^−1^])	m	n	E (kJ/mol)	ln [Z] (ln[min^−1^])	ΔH_TOT_ (J/g)
313	−10.91	0.027	0.717	62.92	13.14	1026.75
316	−11.12	0.113	0.696			994.43
320	−10.15	0.158	1.564			4485.45
323	−10.43	0.096	0.985			6736.28

## Data Availability

All data generated or analyzed during this study are included in this published article.

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
