# Peer review of "RIFT Process Analysis for the Production of Green Composites in Flax Fibers and Bio-Based Epoxy Resin"

_materials, 2022, doi:10.3390/ma15228173_

Round 1

Reviewer 1 Report

Abstract is too general, and does not convey the new findings of this work. The author should rewrite the abstract (try to omit the first two or three sentences which are only the explanation of composite materials).

Significant English editing service is required.

Numerical and experimental results dismatch. For example, the flow front is generally parabolic due to the friction at each side wall, however, the numerical results do not show any parabolicity at the flow front. This fails to appropriate numerical simulations. What's the boundary condition at each side? 

If the boundary condition is non-slip at each side, there should be parabolic line of the flow front.

Mesh creation is too rough and clumsy. Since the simulation region is rectangular, the author can do improve the mesh quality.

Most of all references are old-dated.

What's the viscosity curve of the materials used for resin impregnation? The author should present it since the viscosity involves the Darcy's law and also the numerical results significantly.

Reviewer 2 Report

The study addresses the resin flow simulation using a commercial tool for natural fiber preforms. Unfortunately, the study does not account for the special properties of natural fibers such as liquid absorption and fiber swell. Also, the study seems to be very basic and has been previously addressed by several works (mainly Nguyen et al./ CST/2015).  Also, the premise of the work should be well constructed. I was not able to exactly understand the novelty of this work. 

Futhremore the difference in the fill times for different resins is simply because of their different viscosities and it has nothing to do with the "green" composites. Also, the biobased resin used in this study is primarily used for bonding applications where the viscosity should be high, hence it is normal to see such results. 

Reviewer 3 Report

The publication covers the experimental and numerical method of producing composites such as bio-epoxy resin-linen fabric with the use of the RIFT method. The publication is interesting, but I have a few questions / suggestions:

1. In the context of a broader description of the permeability and porosity of fibers, I propose to enrich the literary introduction to explain the properties and morphology of flax fibers.

2. In order to improve the literature introduction, I recommend providing an overview of the current state of knowledge related to the simulation of the process of supersaturation of fabrics / mats with resin.

3. Natural fibers, including flax fibers, are characterized by a fairly wide dispersion of processing and mechanical properties. How many attempts / tests have been undertaken as part of the work carried out? Please take it into account in terms of statistics when presenting the results, including graphs.

4. Also recently, there have been publications in the literature related to the possibility of predicting the mechanical properties of resin-flax composites using the DIGIMAT software. This is an interesting and important issue, as natural fabrics (such as flax fabric) are characterized by a lack of homogeneity of properties. You can write a few sentences on this topic in the discussion or in the literature introduction.

5. Has the thickness of the laminates been taken into account in the numerical analyzes? Has the porosity of the individual fibers been taken into account?

6. In the materials and methods section, there should be a paragraph on the materials used in the tests in terms of their properties. Some of this information is in the article but scattered throughout the manuscript. What are the properties of the bio-resin used?

7. What data has been entered into the process simulation program? It would be good to present this clearly in the table.

8. There is no deeper discussion related to the obtained results and their comparison with the current state of knowledge and research of other scientists.

9. Has an attempt been made to simulate using other programs, eg Autodesk Moldflow Insight?

10. Please insert in the citations the safety data sheet of the fabrics and resins used.

11.   I have the impression of the logistic chaos of the publication layout. The manuscript should be prepared in accordance with the MDPI recommendations.

Please, explain the above-mentioned questions / comments and correct the manuscript. Good luck!

Reviewer 4 Report

The authors present a study on composites made from natural fibers and a bio-based epoxy resin. The study aims to provide insights on parameters such as compressibility and permeability of the fabric and the curing behavior of the bio-based resin, which are evaluated experimentally. Subsequently, the determined parameters are used in an numerical model.

General comments:

Throughout the manuscript terms like "environmental sustainibility" and "green composites" are used without proper definition. It is unclear to the reader, if these terms apply only for bio-based or bio-degradable polymers or for both and it should be considered that even fully bio-based composites can end up as landfill waste. Hence, they may be "environmentally sustainable" from the sourcing-perspective, but not from the end-of-life-perspective. As a result, the authors should define these terms in the context of their paper.

Introduction:

Line 29: The environmental impact on what is meant?

Line 35: In my perspective natural fiber reinforced composites are not a 1:1 alternative to glass, but can rather compete or excel in distinct applications.

Line 39/40: There are also composite parts containing natural fibers, which are made from nowovens, e.g. with hybrid fiber systems by compression molding.

Line 41/42: Please insert a paragraph before "Usually...", because the topic changes from "green composites" to composite permeability.

Line 52-55: It is unclear what the meaning and/or value of citation 8 is.

Line 55: Please insert a paragraph before "Pierre-....", because the topic changes from "green composites" to composite permeability.

Line 60-62: Please add the resoning for the different approaches for determining in-plane and thickness saturated flow permeability.

Line 67/68: Please add the variation, which was the basis for the increase in permeability and mechanical properties.

Line 77-80: Do I understand the reasoning correct: "Previous publications focused on the permeability of the natural fiber semi-finished-products. Hence, this study focuses on the processing." ? This was unclear to me as this study also focuses on the permeability.

Lin 86/87: Please check grammar and context to the rest of the sentence.

Materials and methods:

Materials

Please add a section where the used materials such as flax fabric and bio-based resin are described in detail. Especially, on the bio-based resin almost no data is available.

Were the same flax fabrics used for the compression and permeability tests?

Methods

A schematic drawing would be very helpful to understand the experimental setup, especially regarding marker positions and dimensions, etc.

Which permeability k was used in the numerical model? It is not clear to me if only a permeability in flow direction or also perpendicular to the flow direction was determined and used in the simulations. Also, wasn't k determined based on vaseline oil? How are these findings transferred to the actual resin system?

Results and discussion:

Line 219/220: "respect" may not be the correct term. Please check grammar and context to the rest of the sentence.

Line 244/245 and later in the conclusion: The viscosity of the resin in infusion processes is a crucial material property. However, no viscosity measurements of the bio-based epoxy are presented and no source for viscosity data of this resin is provided. For comparison: For the vaseline oil the viscosity is stated (line 116) and for the petrochemical resin it is taken from a data sheet (line 319/320). In line 244/245 the authors state that the viscosity of bio-based resins is generally higher without providing a citation or data. The statement is also very general without providing evidence. Since, the authors conclude, that the cavity filling of the bio-based resin was 4 times higher, it is irritating that this finding was not correlated with a viscosity measurment. Hence, it would be very helpful to add viscosity data of the bio-based resin and to discuss the results in this context as well.

Line 323 states: "The numerical results of the case study showed very different filling times." How did the results compare to the experiments with the two resins?

Line 345-348 state: "The bibliographical introduction of green composite materials has highlighted the presence of several works focused on the use of natural fibers but at the same time has highlighted the lack of work on the use of organic resins to ensure a complete environmental sustainability." In my view, this is not clearly shown by the literature review in the "bibliographical introduction". The introduction lists papers regarding "green composites" without a definition of the term and does not hold in my opinion for "the lack of work on the use of organic resins to ensure a complete environmental sustainability." as this is not clearly worked out in the literature review.

Line 358/359: This points back to my comment regarding viscosity of the bio-based resin.

Reviewer 5 Report

The paper is not adding significant new fin to the previously published work. Some of the parts such as selection of epoxy resin seems quite random. What was the criteria applied in selection of the grades? Only physical properties or similar chemical structure? The nature of the resin itself such as molecular weight and functionalization will affect the results.

Three readings for mechanical properties is barely the minimum for any sort of statistical analysis and a larger sample group is needed.

How long is the curing time of epoxy resins used? Is the difference of 200s actually significant in consideration of the setting time for epoxy since the application temperature was ambient room at 25? Was the difference in filling time due to the difference in viscosity of the resins used? 

From industrial application point of view, what issue is this paper addressing in RIFT process? 

There are lots of question marks that makes this paper not suitable for publication in the current state.

Reviewer 6 Report

The manuscript appears to be mathematically and experimentally correct and the utilized methods and the proposed models are interesting. I recommend the paper for publication, however, there are some concerns, comments and suggestion should be addressed before publication:

1.      The following part should be removed from abstract and the abstract should be just about your own work and structure and results of that. “The use of composite materials is increasingly widespread due to the performance related to the considerable weight reduction and the high mechanical properties, such as high stiffness and mechanical strength. On the basis of the current requirements of environmental sustainability, how ever, it is necessary to ensure that these materials are also environmentally sustainable, i.e. "green", and this is possible thanks to the use of natural fibres and bio-based resins. These materials, still 16 little known, have some critical issues related to the lack of knowledge of the fundamental repre sentative parameters for a correct design of the forming process.”

2.     There are grammar and typographic errors. Please correct these errors and further improve the language.

3.     The novelty of this work must be presented in Introduction and Conclusion section very clear.

4.     In the Introduction section, it is suggested that some literature published in recent two years (2020-2021) should be included and also, some of the following literature concerned missing and these should be appropriately cited.

Steel and Composite Structures 35 (5), 659-670, 2020; Sustainability 12 (18), 7683, 2020; The European Physical Journal Plus 136 (6), 646, 2021; Journal of Polymers and the Environment 28, 3029-3054, 2020; Polymer Composites 42, 3508–3517, 2021.

The results and figures are appropriate however; author should add more physical explanation for the observed results.

5.     What is the criterion to determine the occurrence of mechanical behavior?

6.     How to verify the accuracy and correctness of the developed formulation?

Round 2

Reviewer 1 Report

The authors have replied all the concerns and comments.

Author Response

The authors thank the reviewer for his contribution to the improvement of their work.

Reviewer 3 Report

The authors responded and revised the manuscript according to my recommendations and comments.

Author Response

(The authors gave the same response as above.)

Reviewer 4 Report

The author improved the manuscript and provided the necessary background context/information.

Author Response

(The authors gave the same response as above.)

Reviewer 5 Report

Overall, the paper has improved and the authors managed to answer few of the points raised previously. However, in the newly added parts I noticed several grammatical mistakes which needs to be corrected. I list a couple of them below:

1- Line 119 "The choose of this epoxy system was due to the ..." should change to the " The Choice of ..."

2- Line 105 "The objective of this work consists of in studying" which clearly contains extra proposition. 

3-Line 71 "Developments of such models include the implementation of  thermo-chemical models for considering the cure kinetics and the.. " Developments of such models include the implementation of  thermo-chemical models which take into consideration cure kinetics ...

Author Response

The authors thank the reviewer for his contribution to the improvement of their work. The text has been updates as requested.